# Proton Pump Inhibitors and Fracture Risk: A Review of Current Evidence and Mechanisms Involved

**DOI:** 10.3390/ijerph16091571

**Published:** 2019-05-05

**Authors:** Benjamin Ka Seng Thong, Soelaiman Ima-Nirwana, Kok-Yong Chin

**Affiliations:** Department of Pharmacology, Universiti Kebangsaan Malaysia Medical Centre, Cheras 56000, Malaysia; benjamin6126@live.com.my (B.K.S.T.), imasoel@ppukm.ukm.edu.my (S.I.-N.)

**Keywords:** bone, compression, omeprazole, osteoporosis, pantoprazole

## Abstract

The number of patients with gastroesophageal problems taking proton pump inhibitors (PPIs) is increasing. Several studies suggested a possible association between PPIs and fracture risk, especially hip fractures, but the relationship remains contentious. This review aimed to investigate the longitudinal studies published in the last five years on the relationship between PPIs and fracture risk. The mechanism underlying this relationship was also explored. Overall, PPIs were positively associated with elevated fracture risk in multiple studies (*n* = 14), although some studies reported no significant relationship (*n* = 4). Increased gastrin production and hypochlorhydria are the two main mechanisms that affect bone remodeling, mineral absorption, and muscle strength, contributing to increased fracture risk among PPI users. As a conclusion, there is a potential relationship between PPIs and fracture risks. Therefore, patients on long-term PPI treatment should pay attention to bone health status and consider prophylaxis to decrease fracture risk.

## 1. Introduction

Proton pump inhibitors (PPIs) are lipophilic weak bases (pK_a_ 4.0–5.0) consisting of two components: a substituted pyridine and a benzimidazole. PPIs are synthesized as prodrugs and are converted to sulfenic acids or sulfenamides that bind to one or more cysteines of the gastric H^+^/K^+^-ATPase covalently in an acidic environment. There are several subtypes of proton pump inhibitors available in the market, such as esomeprazole, omeprazole, pantoprazole, lansoprazole, rabeprazole, ilaprazole, and dexlansoprazole. Although these PPIs have the same basis structure, they differ in pharmacokinetic and pharmacodynamic profile [1].

In clinical practice, PPIs are commonly used as acid-suppressive agents to treat multiple acid-related gastrointestinal disorders, such as peptic ulcer disease, *Helicobacter pylori* infection, dyspepsia, gastroesophageal reflux disease (GERD), and Zollinger Ellison syndrome [2,3,4,5,6]. Moreover, they also serve as prophylactic agents among users of non-steroidal anti-inflammatory drugs (NSAIDs) to prevent gastric ulcers and bleeding [7,8,9]. Overall, PPIs account for 95% of acid-suppressing drugs prescription due to their effectiveness [10]. In Australia, PPIs are the third most prescribed medications, equivalent to 6.9 million prescriptions in 2014 [11]. In the United States, there were 14.9 million patients receiving 157 million prescriptions of PPIs in the year 2012 [12]. Other than that, PPIs were one of the most frequently sold over-the-counter drug with $13 billion of global market value [13]. Long-term PPI therapy is reported to be associated with decreased bone mineral density (BMD) [14,15,16]. Reduction in BMD and deterioration of bone microstructure are characteristics of osteoporosis, a metabolic bone disease [17]. Osteoporosis ultimately leads to decreased bone strength and susceptibility to fracture [17]. Osteoporotic fracture is becoming a major health concern in the rapidly aging society, owing to the tremendous economic burden, mortality, and morbidity associated with it [18,19,20,21]. In fact, hip fracture is considered as the most common cause of elevated mortality and dependency in elderly patients. Approximately 1.6 million hip fracture cases are reported worldwide each year and the number will reach between 4.5 and 6.3 million by 2050 [20,22,23]. The risk of mortality in hip fracture patients is threefold higher than the general population [24]. In the United States, the estimated age-weighted and lifetime savings for surgical treatment of hip fractures was more than United States dollar (USD) 65,000 for a patient [25]. However, the extent of the problem of PPI-induced osteoporotic fracture is less known.

There were several reviews on the relationship between PPI and fractures in the past [26,27,28,29], but they do not explore the mechanism underlying this relationship. Multiple longitudinal observational studies concluded in the recent five years shed new light on the relationship between PPIs and fracture risk. Therefore, this narrative review aimed to discuss the recent epidemiological findings pertaining to this topic and the possible underlying bone-weakening mechanism of PPIs. In reviewing the relationship between PPI and fracture risk, a literature search was performed using the keywords (“proton pump inhibitors” OR “*prazole” OR “acid suppress*”) AND (fracture OR bone) to identify original research articles indexed in PubMed, Scopus, and Web of Science. Only articles written in English, published between 2013–2018 were included.

## 2. The Relationship between PPIs and Fracture Risk

### 2.1. General Population

Fracture risk in the general population using PPIs, such as children, young adults, men, elderly, and postmenopausal women, was explored in several studies [30,31,32,33,34,35,36,37,38,39,40,41,42]. In the case-control study of Freedberg et al. involving children (4–17 years; *n* = 87,071; 69.8% were cases) and young adults (18–29 years; *n* = 37,728; 30.2% were cases) in the United States (follow-up period: five years), a positive relationship between fracture risk and PPI exposure in young adults was found, with a body mass index (BMI)-adjusted odds ratio (aOR) of 1.39 (95% confidence interval (CI): 1.26–1.53) [41]. However, there was no significant relationship between PPI use and fracture risk in children in terms of cumulative exposure manner [41]. A dose-dependent effect of PPIs on fracture risk was suggested, whereby the effect was only observed in young adults but not in children [41]. The most common fracture sites were wrist (24.9%) and hand (20.5%) for children, and foot (12.4%) and hand (32.5%) for young adults [41]. However, the authors could not identify the etiology of the fracture, and the analysis was not stratified according to ethnicity [41].

In another study examining the positive relationship between PPIs and hip fracture risk among men of diverse ethnic background (70% non-Hispanic Caucasians) in the United States, the subjects were stratified based on age into 45–59 years (*n* = 999), 60–69 years (*n* = 1098), 70–79 years (*n* = 1969), and 80+ years (*n* = 2708) [32]. Subjects with the most recent use of omeprazole (aOR: 1.22; 95% CI: 1.02–1.47) and those with medication possession ratio (MPR) ≥80% (aOR: 1.33; 95% CI: 1.09–1.62) had an elevated risk of hip fracture [32]. Moreover, the authors suggested that patients on omeprazole for more than 2.5 months had an increased hip fracture risk [32]. They also demonstrated that cumulative exposure, frequency exposure, and duration of PPI use influenced fracture risk of the patients [32]. However, pantoprazole use was not significantly related to hip fracture risk after adjusting for other comorbidities [32]. This study benefited from the availability of electronic medical records in collecting data on medication use and fracture incidence. A limitation of note was that omeprazole and pantoprazole were the only generic ingredients investigated.

Several studies focused on the relationship between PPIs and fracture among postmenopausal women [36,40,42]. A retrospective study conducted among Australian women (*n* = 1045, mean age = 76.5 ± 2.6 for non-users and mean age = 77.5 ± 2.5 for users) reported an increased fracture risk in patients taking PPIs ≥1 year or ≥1.5 standard daily doses [42]. Women with long-term PPI were two times more likely to experience a fracture (aOR: 2.07 (95% CI: 1.13–3.77)) [42]. Moreover, a significantly lower serum vitamin B_12_ level in PPI users was detected. This prompted the researchers to suggest that patients on high-dose and longer-term PPI had an increased fall risk [42]. Additionally, an increase in fracture risk among postmenopausal women (*n* = 6917, mean age = 56.4, follow-up period = 14.4 years) using PPI was also reported by a Swedish retrospective study (aOR: 2.01 (95% CI: 1.31–3.08)) [40]. Moreover, PPI use was also reported to increase fracture risk in Australian elderly women (*n* = 1396, mean age = 78.2 years for PPI users; *n* = 1338, mean age = 78.3 years) [36]. Only patients on rabeprazole and multiple types of PPI were found to have increased fracture risk [36].

The relationship between PPIs and fracture risk in the elderly is debatable [30,37,39]. A data mining study utilizing the Food and Drug Administration (FDA) adverse event reporting system investigated 169,562 entries with PPI use [38]. The mean age of the PPI users with fracture was 65.3 years and a sex ratio (female to male) of 3.4. The fracture sites reported by PPI users were the thoracic cage and pelvic bone [38]. Ding et al. conducted a large retrospective cohort study (*n* = 25,276) with a follow-up period of five years using data from the Pharmaceutical Assistance Contract for the Elderly (PACE) program [39]. PPI users were 1.27 times more likely to experience a fracture, including major osteoporotic fracture, hip fracture, vertebral fracture, and other fractures compared to non-users [39]. In another cohort study, there was a significant increase in fracture risk (adjusted hazard ratio: 1.40; 95% CI: 1.11–1.77) among PPI-users (mean age of 67.6 years) after using PPIs for a decade [34]. However, hip fracture risk did not elevate significantly in their study [34]. On the other hand, Lee et al. discovered a positive relationship between hip fracture risk and PPI use (aOR: 1.34 (95% CI: 1.24–1.44)) among Korean population (*n* = 24,710 for cases and *n* = 98,642 for controls; mean age = 77.7) [43]. Interestingly, there was a positive dose relationship among PPI users who used bisphosphonates concurrently but not in subjects using PPIs only [43]. Furthermore, when the subjects were stratified based on drugs, a significantly increased hip fracture risk in populations using pantoprazole, rabeprazole, and omeprazole was observed [43]. Soeriano et al. performed a nested case-control study in a United Kingdom (UK) population aged 40–89 years (*n* = 10,958 for cases; *n* = 20,000 for control) and found that single PPI use elevated hip fracture risk (aOR: 1.09 (95% CI: 1.01–1.17)) [35]. Further sub-analysis revealed that the relationship was dose-dependent but not time-dependent [35]. In this study, the subjects included were already at high risk for fracture, as the majority were women aged ≥60 years old [35].

In contrast, a study involving a Taiwanese elderly population using PPI (*n* = 14,416; mean age = 79.8 years for cases, mean age = 79.7 for control) observed that PPI exposure did not increase hip fracture risk [30]. The relationship was not significant in terms of cumulative duration, cumulative dose or recentness of use [30]. Furthermore, a retrospective cohort by Harding et al. (follow-up period: 6.1 years) found similar results among older adults (*n* = 4438) with a median age of 74.0 years in the United States, whereby PPIs did not elevate fracture risk in a dose-, age-, and sex-dependent manner [37]. The most common fracture site in this study was hip (23%) followed by forearm (22%), rib and sternum (18%), humerus (13%), ankle (9%), pelvis (6%), tibia and fibula (6%), and clavicle and scapula (3%) [37]. In the Mediterranean elderly populations (*n* = 1056, mean age = 82, SD = 8.8), PPIs, regardless of the duration of exposure, type, or dose, were not associated with increased hip fracture risk [31].

### 2.2. Patients with Comorbidities

The risk of fractures in association with PPIs in patients with specific conditions, such as stroke, GERD, Alzheimer’s, hemodialysis, and kidney transplant was also studied [33,44,45,46,47]. Two studies performed in Taiwan used the National Health Insurance Research Database to examine the association between PPIs and fracture risk in GERD and stroke patients [33,47]. A retrospective cohort study with a total of 31,358 Taiwanese GERD patients (mean age: 46.6 for control and 47.1 for GERD patients; follow-up period = 3.45 years for GERD and 3.55 years for control) suggested that there was no relationship between PPI and hip fracture risk, despite a significant association between osteoporosis and PPI use [33]. This was a large population study, but the diagnosis of osteoporosis was not validated with densitometry results [33].

In another study using the same database, the researchers included 5298 entries of Taiwanese stroke patients with the mean age of 66.7 and 66.9 for PPI users and non-PPI users, respectively [47]. The study identified that patients using PPI were more likely to suffer from a hip fracture (adjusted hazard ratio (aHR): 1.18; 95% CI: 1.00–1.38) and a vertebral fracture (aHR: 1.33; 95% CI: 1.14–1.54) after a mean follow-up period of 4.8 years. The relationship was dependent on dose and cumulative exposure [47]. Similar to the previous study, a significant association between osteoporosis and PPI use was found [33,47]. The limitation of using the National Health Insurance Research Database was that the patients’ lifestyle, physical, and medical data could not be reviewed [33,47]. Therefore, the effects of potential confounding variables could not be adjusted.

In Alzheimer’s disease patients with a higher fall risk, a nested case-control study in Finland (*n* = 4818, mean age = 84.1 for cases; *n* = 19,235, mean age = 84.0 for controls) demonstrated that the use of PPI modestly increased the risk of hip fracture among short-term PPI current users (<1 year) [44]. However, long-term or cumulative PPI users did not have a significantly higher hip fracture risk [44]. This study reported a reliable documentation system on drug use period and the use of representative nationwide data [44]. On the contrary, the lack of serum cobalamin data was acknowledged [44]. The reason given as to why short-term PPI use affected bone was that the current PPI users in the study had more comorbidities. The researchers claimed that, although they adjusted the analysis to account for the comorbidities, the effects of the underlying health conditions influencing the use of PPI might not have been fully captured.

Furthermore, two studies in the United States involving patients with renal issues found a positive relationship between PPI use and fracture risk [45,46]. Vangala et al. performed a retrospective case-control study on hemodialysis patients (*n* = 4551, mean age = 71 ± 12 for cases, and *n* = 45,510, mean age = 61 ± 14 for controls) and found that those exposed to PPI showed increased hip fracture risk (aOR: 1.19; 95% CI: 1.11–1.28) [45]. In terms of cumulative exposure, patients who took PPI < 20%, 20% ≤ *x* < 80%, and ≥80% of the time for three years were all at risk for hip fracture [45]. Similarly, another study showed a positive relationship between PPI and fracture risk in kidney transplant recipients (*n* = 231, mean age = 51.8 ± 12.9 for cases, and *n* = 15,575, mean age = 51.2 ± 10.4 for control) but a clear frequency-dependent relationship was not observed [46]. Their study included a large sample size of kidney transplant recipients, but they did not investigate the dose and the type of PPI used [46]. A meta-analysis with 33 studies and a combined sample size of 2,714,502 subjects (33.21% men, mean age 66.91 years (95% CI: 63.37–70.46)) confirmed that PPI exposure might increase the risk of fractures but exerted no effect on BMD. From the analysis, the fracture incidence of PPI users was 22.04% (95% CI: 16.10–27.97) and 15.57% (95% CI: 12.28–18.86) for non-users. The OR also increased with the duration of use. Table 1 summarizes the findings on the relationship between PPIs and fracture risk.

## 3. Mechanism of Bone Fractures induced by PPI

Two main mechanisms underlying bone fractures induced by PPIs, i.e., hypergastrinemia and hypochlorhydria, are discussed in this section (Figure 1).

### 3.1. Hormones and PPIs

Hypergastrinemia plays a role in altering bone metabolism. Due to the irreversible binding of PPI to H^+^/K^+^-ATPase of parietal cells, the concentration of hydrogen ions secreted in the stomach is reduced, resulting in increased gastric pH [48]. The elevated pH, in turn, suppresses the secretion of somatostatin from mucosal D-cells positioned in the gastric antrum. As a result, the G-cells are activated to secrete gastrin to stimulate the parietal cells and enterochromaffin-like cells (ECLCs) to release hydrogen ions and histamine, respectively [49,50,51]. The ECLCs produce histamine to further stimulate parietal cells [52]. However, due to the prolonged action of PPI in the stomach, the G-cells oversecrete gastrin and lead to hypergastrinemia, resulting in ECLC hyperplasia and increased secretion of histamine by the ECLCs [53,54,55,56,57,58,59].

Multiple studies showed the relationship between PPI and hypergastrinemia [53,54,55,58,60]. A cellular study showed that 10^−5^ M rabeprazole induced a significantly elevated histamine secretion compared to the basal level [57]. At basal conditions, ECLCs only secreted 7% of the intracellular histamine content. With the stimulation by gastrin and rabeprazole, the secretion of intracellular histamine increased to 24% and 11%, respectively [57]. Wistar rats treated with 30 mg of rabeprazole for two weeks showed a 1.8-fold increase in the concentration of serum gastrin and a 3.9-fold increase in histidine decarboxylase activity [55]. The histidine decarboxylase functions to increase the histamine secretion and is positively linked with PPI use and histamine concentration [61,62]. In a randomized control trial by Arroyo et al., 40 duodenal ulcer patients were assigned to take either 20 mg of omeprazole per day or 30 mg of lansoprazole per day [53]. Serum gastrin was measured every two months, and they found mild hypergastrinemia in these patients. In another double-blind randomized control trial, rabeprazole (10 mg/20 mg) or omeprazole (20 mg) was prescribed to patients for five years (*n* = 243, 51% completed the trial), and a strong association between concentration of serum gastrin and ECLC hyperplasia (*p* = 0.001 for gastrin effect) was found [58]. Moreover, medium and high serum gastrin concentrations were predictive of ECLC hyperplasia (medium level = aOR: 3.43; 95% CI: 1.58–7.46; high level = aOR: 4.53; 95% CI: 1.69–12.22) [58]. Additionally, a Canadian cohort study with mean follow-up period of 2.29 years concluded that long-term PPIs and infection by *H. pylori* were risk factors for ECLC hyperplasia [60]. Furthermore, patients who received PPI treatment for >6 months were found to have a mean serum gastrin level of 125.67 pg/mL (68–267), which was higher than the control group [60].

Since histamine could increase the differentiation of osteoclast precursors [63], the effects of hypersecretion of histamine due to ECLC hyperplasia on bone should be a concern. However, to date, there is no study linking the effect of increased circulating histamine concentration on bone. Some researchers doubt that the low circulating level of histamine would pose systemic effects [57]. Therefore, further research should be carried out to validate this hypothesis.

Apart from histamine, hypergastrinemia is related to hyperparathyroidism [48,64,65,66,67,68]. Chickens receiving five weeks of omeprazole treatment developed hypergastrinemia, as well as hyperplasia and hypertrophy of the parathyroid glands, associated with increased parathyroid hormone (PTH) gene expression [64,65]. The femur density of the chickens also reduced significantly [65]. These changes on PTH glands could not be averted with ergocalciferol supplementation [65]. In addition, rats with hypergastrinemia induced by antral exclusion also developed hyperparathyroidism and increased parathyroid gland volume and weight due to hyperplasia of the parenchymal cells [67]. However, evidence on the relationship between hypergastrinemia and hyperparathyroidism in humans is more heterogenous [48,66,69,70]. In a double-blind and two-period cross-over study by Dammann et al., 12 healthy young men were treated with oral rabeprazole (20 mg per day for 14 days), but no clinically relevant effect on T3, T4, and PTH level was observed [48]. However, this study was limited by its small sample size and short duration. In contrast, a retrospective study performed in the United States among 80 subjects aged ≥60 years old demonstrated that chronic PPI users had a significantly higher PTH level (65.5 vs. 30.3 pg/mL; normal range 10–55 pg/mL) and a lower serum calcium level (9.1 vs. 9.4 mg/dL; normal range 8.5–10.5 mg/dL) compared to non-users [69]. A Japanese study showed that bone turnover was affected by hyperparathyroidism induced by PPI. Exposure of PPI among Japanese patients with gastric ulcer (*n* = 19; aged 67 ± 13 years) led to a 28% increase in the mean PTH level and serum bone turnover markers (serum osteocalcin and alkaline phosphatase), and a decrease in the renal excretion of calcium and hydroxyproline [66]. The evidence from this study was preliminary due to several limitations, such as single time-point PTH reading and the lack of serum gastrin results. Therefore, further investigation on the relationship of PPIs and PTH should be conducted. On the other hand, parathyroid hormone-related peptide (PTHrP) was shown to be regulated by gastrin [71]. This growth factor is expressed in acid-secreting parietal cells of the mouse stomach, and gastrin could activate its expression via transcription activation and messenger RNA (mRNA) stabilization [71]. Since PTHrP is related to bone metabolism and chondrogenesis [72,73,74,75,76,77,78], the relationship between PPI-induced PTHrP and bone metabolism should be examined.

### 3.2. Micronutrients and PPIs

Hypochlorhydria caused by PPIs is hypothesized to decrease the absorption of minerals essential for bone health. Calcium is indispensable in maintaining bone microstructure [79,80]. However, prolonged hypochlorhydria and increased gastric pH due to PPIs could reduce ionization of calcium and affect the intestinal absorption. The reduction in circulating calcium promotes the parathyroid gland to secrete more PTH to mobilize calcium storage in the bone [81,82]. This process, which involves bone resorption, deteriorates the bone microstructure and strength, and increases the fracture risk [83]. In a randomized, double-blind, placebo-controlled, cross-over trial by O’Connell et al. involving 18 women (mean age of 76 ± 7 years), 20 mg of omeprazole significantly reduced fractional calcium absorption by 9.1% (95% CI: 6.5–11.6%) compared to 3.5% (95% CI: 1.6–5.5%) caused by placebo [84]. This would explain the reduction in BMD in patients on long-term PPI therapy as observed in multiple human studies [14,15,16]. They included 18 women with a mean age of 76 ± 7 years and a mean weight of 61 ± 7 kg [84]. This mechanism remains controversial and further study is needed.

Furthermore, the higher gastric pH is suggested to decrease intestinal absorption of magnesium. Magnesium deficiency is detrimental to bone by inducing formation of large hydroxyapatite crystals which decrease bone stiffness, decrease osteoblastic activity, and increase the number of osteoclasts, as well as promoting inflammation and oxidative stress [85]. However, observations from human epidemiological studies are heterogeneous. A mixed-design study demonstrated clinically significant hypomagnesemia among patients on PPI therapy (*n* = 100) ranging from <1 to >5 years of duration [86]. They also indicated that serum magnesium levels in new PPI users (*n* = 56) declined with time [86]. A meta-analysis of nine studies with 115,455 patients by Cegla et al. demonstrated a risk of hypomagnesemia among patients using PPIs (pooled OR: 1.775; 95% CI: 1.077–2.924) despite significant heterogeneity of the studies included [87]. Since hypomagnesemia is not readily identifiable on regular blood testing, many patients might develop this condition unknowingly. In contrast, data of 2400 patients were retrospectively inspected by Chowdhry et al. and they did not find significant differences between the PPI users and non-users [88]. The lack of significance persisted even after considering the dose of PPI and concomitant use of diuretics [88]. This was confirmed by another prospective study, whereby PPI treatment for 12 months did not lower serum magnesium level significantly among patients (*n* = 209) [89].

Some researchers postulated that renal problems could potentiate the magnesium-lowering effect of PPI. Hughes et al. revealed that chronic kidney disease (CKD) patients (*n* = 1230) using PPI, regardless of drug type, developed hypomagnesemia (aOR: 1.12; 95% CI: 1.06–1.18) [90]. PPI users undergoing chronic hemodialysis (*n* = 170; 0.94 ± 0.2 mmol/L) were reported to have significantly lower serum magnesium levels compared to non-users (*n* = 112; 1.03 ± 0.2 mmol/L) regardless of duration of treatment [91]. In the same study, multivariate analysis confirmed that the use of PPIs was a strong predictor of low magnesium concentration (OR 3.05; 95% CI: 1.2498–7.4594, *p* = 0.01) [91]. In contrast, Erdem et al. showed that, in hemodialysis patients, serum magnesium levels were predicted by the patient’s dialysate magnesium concentration and not by PPI use [92]. In addition, there was no significant difference between serum magnesium levels between patients using PPIs (2.73 ± 0.3 mg/dL) and non-users (2.88 ± 0.3 mg/dL) [92].

Other studies found that the absorption of vitamin B was negatively affected by PPIs [93,94]. Vitamin B_6_, folate, and vitamin B_12_ deficiencies lead to the development of hyperhomocysteinemia, which in turn affects bone quality. A high level of homocysteine will inhibit the enzyme lysyl oxidase and affect formation of crosslink in collagen fibers. This will reduce the quality of the skeletal collagenous matrix [95]. Moreover, suboptimal vitamin B level increases the risk of neurological diseases and contributes to muscle weakness, which will increase the risk of falls and fractures [96,97,98]. A recent case-control study showed that vitamin B_12_ deficiency was associated with chronic current use of acid-suppressive agents such as PPIs and H2-receptor blockers (aOR: 4.45; 95% CI: 1.47–13.34), but not past or short-term current use of these agents [94]. Similarly, a clinical trial by Marcuard et al. revealed that, after a two-week 20-mg omeprazole therapy, the cyanocobalamin (vitamin B_12_) absorption was found to be significantly decreased from 3.2% to 0.9% in healthy men [99]. In participants receiving 40 mg of omeprazole daily, the cyanocobalamin absorption decreased from 3.4% to 0.4% (*p* < 0.05) [99]. Therefore, they concluded that the decreased absorption of vitamin B_12_ was in a dose-dependent manner [99]. On the contrary, a recent retrospective study indicated no association between PPI use and impaired vitamin B_12_ users (*n* = 658; mean age of 73 ± 16 years) [93]. This study was confounded by the fact that half of the PPI users were using multivitamin concurrently, which normalized their vitamin B_12_ levels. A meta-analysis conducted by Jung et al. with four case-control studies (4254 cases and 19,228 controls) demonstrated that long-term use of acid-lowering agents was significantly associated with vitamin B_12_ deficiency (aHR: 1.83; 95% CI: 1.36–2.46) [100]. Although vitamin B_2_, B_6_, and folate are reported to be associated with bone health, there is a lack of studies that illustrate the relationship between PPIs and these vitamin B subtypes. Additionally, evidence on the relationship between vitamin B_2_, B_6_, and folate and bone health remains heterogenous in human studies [101]. A large study among the Chinese population (*n* = 63,154; mean follow-up: 13.8 years) found no significant effect of vitamin B_3_ on bone [102].

The relationship between PPI use and fall risk was validated in a meta-analysis involving 367,068 patients, which demonstrated that the relationship between PPIs and fall risk was significant (OR: 1.27; 95% CI: 1.07–1.50) [103]. However, there is a paucity of studies which consider the interrelation among PPIs, vitamin B level, and fracture risk together [95,101,104]. This aspect should be scrutinized in the future to resolve the complexities between these factors and fracture risk. Studies suggested vitamin B_12_ deficiency can lead to the development of peripheral neuropathy [105]. The demyelination in peripheral nerve can contribute to muscle weakness, functional disabilities, impaired balance, and subsequently increased risk of fall and bone fracture [106,107,108,109]. However, the associations between vitamin B_12_ deficiency, neurological function, and fall are a complex subject, beyond the scope of this review. The linkage between PPI use, vitamin B level, and risk of fall/fracture as hypothesized here remains speculative until proven by large epidemiological studies.

An overview of the systemic effects of PPI contributing to fracture risk is summarized in Figure 1.

### 3.3. Effects of PPIs on Bone Cells

All the systemic effects of PPIs will ultimately influence osteoblasts and osteoclasts, the main players of bone turnover. The osteoblast is responsible for bone formation, while the osteoclast is responsible for bone resorption. The optimal balance between these two biological processes helps maintain bone microstructure [110]. Several studies explored the direct cellular effects of PPIs on osteoclasts and osteoblasts [111,112,113,114] (Figure 2). In osteoblasts, studies suggested that PPI significantly increased osteoblast viability (1–10 µg/mL pantoprazole) and differentiation marked by elevated osteocalcin level (1–4 µg/mL omeprazole) and alkaline phosphatase (ALP) (3–10 µg/mL pantoprazole) activity in MC3T3-E1 cells and human osteoblasts, respectively [111,114]. However, in the study of Costa-Rodrigues et al., a decrease in ALP activity in cultured human mesenchymal stem cells (HMSCs) was noted on day 14 and day 21 when 10^−6^ M–10^−3^ M omeprazole, esomeprazole, and lansoprazole were added [112]. They also noted decreased gene expression of type 1 collagen, ALP, and bone morphogenetic protein-2 in HMSCs treated with PPIs [112].

In peripheral blood mononuclear cell (PBMC) culture, PPIs inhibited the formation of tartrate-resistant acid phosphatase (TRAP)-positive multinucleated cells at day 21 when 10^−5^ M–10^−3^ M omeprazole was used [112]. This was accompanied by a reduction in the markers related to resorptive activity, such as c-myc, c-src, TRAP, and cathepsin K, when the concentration of the PPI was ≥10^−5^ M [112]. The suppression of osteoclast formation was further validated by the reduced expression of osteoclast differentiation markers, such as calcitonin receptor, c-fos, nuclear factor of activated T-cells (NFATc1), and matrix metalloproteinase 9 (MMP9), in an osteoclast precursor (RAW 264.7) culture treated with 0.1–4 µg/mL omeprazole (*p* < 0.001) [111]. However, Hyun et al. showed that there were no significant changes in the number of TRAP^+^ osteoclast-like cells [111]. Omeprazole was noted to increase the ratio of osteoprotegerin to receptor activator of nuclear factor kappa-Β ligand in an osteoblast (MC3T3-E1) culture when 4 µg/mL omeprazole was applied [111]. An overview of the cellular effects of PPIs on bone cells is summarized in Figure 2.

The conflicting results on the effects of PPIs on bone cells in vitro do not help to explain their effects on bone fracture. The number of in vitro studies regarding the direct effects of PPIs on bone cells are limited. The readers should also note that the in vitro studies did not take the skeletal effects of hypergastrinemia, mineral malabsorption, and vitamin B_12_ deficiency induced by PPIs in vivo into consideration. Therefore, the direct effects of PPIs on bone remodeling based on cell culture studies are debatable.

## 4. Conclusions

There are limited studies on the effects of PPIs on bone health in the younger populations. As peak bone mass is an important determinant of bone health, it is important to know whether PPI use will affect peak bone mass attainment and osteoporosis risk later in adulthood. More epidemiological data in Asian countries are required because the skeletal response of Asians toward PPIs might be different from Caucasians. Inclusion of endogenous factors (histamines, PTH, and PTHrP) and dietary components (minerals and vitamin B) in future studies would help to explain the skeletal action of PPIs. Since the direct action of PPIs on bone cells is still elusive, more cellular studies would be helpful to understand its mechanism of action.

The current review suggests that the relationship between long-term PPI use and fracture is still unclear; however, the risk is more apparent in patients with strong secondary risk factor of osteoporosis, such as renal dysfunction. The possible mechanisms of fractures induced by PPIs include hypersecretion of histamine and hyperparathyroidism due to hypergastrinemia, as well as mineral and vitamin B malabsorption due to hypochlorhydria. PPIs may also have direct actions on bone cells, but the studies are limited. Considering the possible burden of fracture, bone health and mineral status of patients on long-term PPIs should be regularly monitored. Routine prophylaxis for osteoporosis is suggested for PPI users to avoid osteoporotic fractures.

## Figures and Tables

**Figure 1 ijerph-16-01571-f001:**
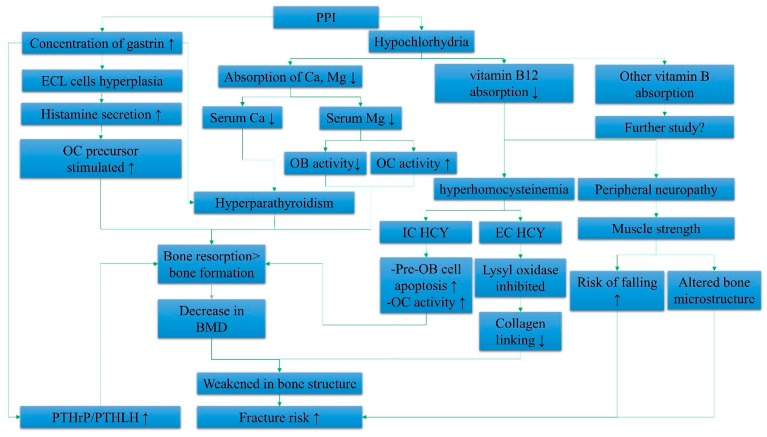
Summary of the systemic effects of proton pump inhibitors (PPIs) in elevating fracture risk. Abbreviations: IC HCY, intracellular homocysteine; EC HCY, extracellular homocysteine; ECL cells, enterochromaffin-like cells; Ca, calcium; Mg, magnesium; BMD, bone mineral density; PTHrP, parathyroid hormone-related peptide; PTHLH, parathyroid hormone-like hormone; OB, osteoblast; OC, osteoclast; ↑, increase; ↓, decrease.

**Figure 2 ijerph-16-01571-f002:**
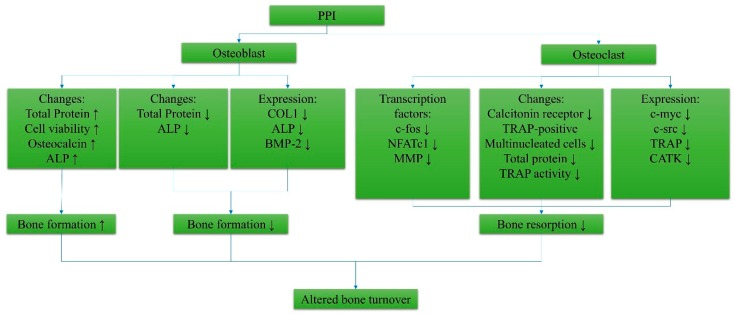
Summary of the cellular effects of proton pump inhibitors on bone cells. Abbreviations: PPI, proton pump inhibitor; ALP, alkaline phosphatase; TRAP, tartrate-resistant acid phosphatase; COL 1, collagen type 1; BMP-2, bone morphogenetic protein-2; NFATc1, nuclear factor of activated T-cells; MMP, matrix metalloproteinase 9; CATK, cathepsin K.

**Table 1 ijerph-16-01571-t001:** Summary of epidemiological studies. PPI—proton pump inhibitor; USA—United States if America; UK—United Kingdom; GERD—gastroesophageal reflux disease.

Reference	Study Design and Population Characteristics	Major Findings
POSITIVE ASSOCIATION		
[34]	Design: population-based cohort, follow-up period: 10 years Subjects: Canadian PPI users (*n* = 261), non-users (*n* = 9162) Mean age in years (SD): PPI = 67.6 (11.1), without PPI = 61.9 (13.4) Gender, %: PPI male (M): 21.8, female (F): 78.2; without PPI M: 30.9, F: 69.1	PPI use increased ten-year but not five-year risk of any fracture. PPI use did not increase hip fracture risk.
[43]	Design: case-control Subjects: Korean elderly ≥65 (mean: 77.7 ± 7.3 for both case and control) Sample: *n* = 24,710 (cases) *n* = 98,642 (control) Gender, %: M case and control: 26.6; F case: 73.5, control: 73.4	Cumulative exposure or ever exposure to PPI increased hip fracture risk. Hip fracture risk increased even the last dose used was 90 days or more prior to index date. The use of pantoprazole, rabeprazole, and omeprazole increased the hip fracture risk, but not esomeprazole and lansoprazole.
[32]	Design: case-controlSubjects: USA men. *n* = 6774 (control) *n* = 6774 (case) Age (for both case, control): 14.7% <60, 85.3% ≥60 Race, %: White (71.4), Black (6.6), Hispanic (11.1), other (10.9) (for both case and control)	Ever use of omeprazole or medication possession ratio (MPR) >80% increased the hip fracture risk in a time-dependent manner. The most recent use (1–7 days) prior to index date (ID) increased the hip fracture risk. Ever use of pantoprazole or any MPR did not increase hip fracture risk. Pantoprazole use for 417–1931 days or if the last dose was 1–33 days prior to ID increased hip fracture risk.
[39]	Design: retrospective cohort study, follow-up (years): 5 Subjects: USA elderly (Caucasians and non-Caucasians) >64 years *n* = 1604 (PPI), *n* = 23,672 (without PPI) Gender, %: PPI: M: 17.1 F: 82.9; without PPI: M: 18.7 F: 81.3	The use of PPI increased the risk of any fractures, major osteoporotic fractures, hip fracture, vertebral fracture and other fractures but not wrist and humerus fracture. For PPI adherence, proportion of days covered (PDC) ≥ 0.80 increased the risk of any fractures, major osteoporotic fracture and other fracture. While for PDC 0.40–0.79, the risk of any fracture also increased.
[42]	Design: prospective cohort study Subjects: Australian elderly woman *n* (without PPI) = 925, *n* (PPI) = 120 Mean age in years: 76.5 ± 2.6 (non), 77.5 ± 2.5 (PPI)	Risk of fracture increased with PPI therapy ≥1 year or ≥1.5 standard daily dose.
[40]	Design: Retrospective cohort, mean follow-up period: 14.4 years Subjects: 6917 Swedish women Mean age in years: 56.4 (50.0–64.0)	The use of PPI increased fracture risk.
[35]	Design: cohort with a nested case-control Subjects: UK men and women aged 40–89 *n* = 10,958 (cases) *n* = 20,000 (control) Age, %: Case: <60: 9, ≥60: 91; Control: <60: 9.7, ≥60: 90.3 Gender, %: case M: 24.8, F: 75.2; control: M: 25.6, F: 74.4	Current use of single type PPI or the last dose of PPI 31–90 days prior to index date (ID) increased hip fracture risk. Medium and high PPI did increase the hip fracture risk. The relationship was not in a time-dependent manner. Only omeprazole increased the hip fracture risk while others such as lansoprazole, pantoprazole, rabeprazole, and esomeprazole did not.
[41]	Design: case-control, mean follow-up period: 5 (SD 3.3) years Subjects: USA children, young adults (*n* = 124,799 cases and 605,643 controls) Age: case, %: <18: 69.8, ≥18: 30.2; Control, %:<18: 70, ≥18: 30 Gender, %: Case M: 65.6, F: 34.4; Control M: 65.4, F: 34.6	Children (<18 years): maximal dose of PPI for daily use or less increased fracture risk but not in a cumulative exposure manner. Young adults (≥18 years): maximal dose of PPI was associated with increased fracture risk in a time-dependent manner.
[36]	Design: prospective cohort, mean follow-up period: 6.6 years Subjects: Australian elderly women *n* (PPI) = 1396, *n* (non-PPI users) = 1338 Mean age in years: 78.2 (1.4) (PPI users) 78.3 (1.5) (non-PPI users)	PPI use increased fracture risk when medication adherence increased. Only rabeprazole and multiple types of PPI were associated with increased fracture risk while others such as omeprazole, lansoprazole, pantoprazole, and esomeprazole were not associated.
[46]	Design: retrospective nested matched case-control, follow-up period: 6.9 ± 5.3 years. Subjects: Kidney transplant recipient (*n* = 231 for case, 15,575 for control) Mean age in years: 51.8 ± 12.9 (case), 51.2 ± 10.4 (control) Gender, %: Case: M: 56.8; Control: M: 49.8 Race, %: African American: 11.3 (case), 7.2 (control) Non-African American Hispanic: 19 (case), 33.4 (control) Missing: 4.3 (case), 1.2 (control)	Use of PPI in ≥80% of the time in one year increased hip fracture risk.
[38]	Design: retrospective study, using Food and Drug Administration Adverse Event Reporting System Data Mining Set with post-marketing surveillance data. Subjects: 169,563 entries in the database with PPI use.	Mean age of PPI users reporting fracture was 65.3 years, and with a gender ratio (F:M) of 3.4:1. Fractures reported to be associated with PPI use included bone sites rich in trabecular bones and atypical bone sites, like rib. PPI use (overall) and 5 generic ingredients (omeprazole, esomeprazole, pantoprazole, lansoprazole, rabeprazole) were reported to be associated with fractures.
[44]	Design: nested case-control Subjects: Finish elderly with Alzheimer’s disease*n* (fracture) = 4818, *n* (control) = 19,235 Mean age in years: 84.1 (cases), 84.0 (control) Gender, %: Both: F: 75, M: 25	Long-term or cumulative PPI use did not increase hip fracture risk, but risk of hip fracture was modestly increased during current short-term PPI use.
[47]	Design: retrospective cohort study mean follow-up period: 4.8 years Subjects: Taiwanese stroke patients *n* = 5298 (For both PPI and non-users) Mean age in years: Yes: 66.7 ± 12.7; No: 66.9 ± 13.1 Gender, %: Yes: M: 62.8, F: 37.2; No: M: 63.7, F: 36.3	PPI use increased risk of hip and vertebral fracture. Cumulative exposure of PPI increased vertebral fracture.
[45]	Design: retrospective case-control Subjects: *n* = 4551 (cases), *n* = 45,510 (controls) Mean age in years: 71 (cases), 61 (controls) Gender, %: case F: 59, M: 41; control F: 52, M: 48	End-stage kidney disease patients on hemodialysis and PPIs were associated with hip fracture events.
NIL RELATIONSHIP		
[31]	Design: Retrospective multicenter case-control Subjects = Spanish elderly 82 (SD = 8.8) (*n* = 358 case, *n* = 698 controls) Mean age in years: 82 (SD = 8.8) (case), 81.9 (control) Gender, %: case F: 77.1, M: 22.9; control F: 76.9, M: 23.1	Continuous/discontinuous exposure of PPI did not increase hip fracture risk. Types and dose of PPI was not related to increased hip fracture risk.
[33]	Design: retrospective cohort, mean follow-up period: 3.45 (GERD) 3.55 (control) Subjects: Taiwanese GERD patients on PPI (*n* = 10,620; men: 56.2%) and control (*n* = 20,738; men: 56%) Mean age in years: 46.6 (SD 14.1); (Control) 47.1 (SD 14.1) (GERD)	GERD patients using PPI did not have an increased hip fracture risk.
[30]	Design: case-control Subjects: Taiwanese elderly, *n* = 7208 Mean age in years: 79.8 ± 7.0 (cases) 79.7 ± 6.9 (control) Gender, %: F: 60.3, M: 39.7 (both)	When the last dose of PPI was ≥12 months prior to index date, hip fracture risk increased. The risk was not related to cumulative duration or cumulative dosage of PPI use.
[37]	Design: retrospective cohort; follow-up period: 6.1 years Sample: 4438 USA men (42%) and women (58%) Age in years: median = 74, IQR = 69.8–79.5 Gender, %: F: 58, M: 42 Race, %: White (89.7), Black (3.8) Asian (3.1), other (3.3)	Use of PPI was not associated with increased risk of fractures.

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
