# Peer review of "Proton Pump Inhibitors and Fracture Risk: A Review of Current Evidence and Mechanisms Involved"

_ijerph, 2019, doi:10.3390/ijerph16091571_

Round 1

Reviewer 1 Report

Dear authors,

For me as Trauma surgeon your article was very interesting to read but I am not that familiar any more with all Details of the bone metabolism. Nethertheless it gave me a nice overview of the recent investigations on Bone-PPI-Interactions.

Author Response

Dear Reviewer, 

Thank you for your time. We value your review and constructive comments. Please kindly find our reply in the attached response sheet. Thank you. 

Reviewer 2 Report

This paper is a very good introduction into the area of PPI use and fracture risk. The boolian operators and key words selected Imply that there was no cherry picking of data. On p2 line 51 there is a small grammatical error. Should read "were included" not "was included."

Author Response

(The authors gave the same response as above.)

Reviewer 3 Report

Dear authors,

The work aimed at presenting and discussing recent epidemiological findings about the chronic use of Proton Pump Inhibitors and bone diseases - fracture risk. This subject is of utmost nowadays, since we observe an increase in PPI uptake.

Nevertheless, I have significant concerns about the relevance and methodology used by this study. Some general aspects are:

1) You do not present a detailed methodology of this systematic review. You only state at the end of Introduction that "a literature search was performed using the 48 keywords (“proton pump inhibitors” OR “*prazole” OR “acid suppress*”) AND (fracture OR bone) 49 to identify original research articles indexed in PubMed, Scopus and Web of Science. Only articles 50 written in English, published between 2013-2018 was included". I would recommend an extensive reading of PRISMA guidelines for systematic reviews (http://www.prisma-statement.org/PRISMAStatement/Default.aspx);

2) In a quick search in the literature, I've found two systematic reviews published in two respected journals this year, about the same subject:

https://www.ncbi.nlm.nih.gov/pubmed/30605646 (Life Sciences)

https://www.ncbi.nlm.nih.gov/pubmed/30539272 (Osteoporosis International)

Therefore, the question is: what's new in this submitted paper? It has to be very clear to IJERPH audience.

Author Response

(The authors gave the same response as above.)

Reviewer 4 Report

General comments:

Based on the information you presented , the conclusion  should state there are   conflicting and mixed data on fracture risk, low bone density  and mechanisms with PPI and bone health.

 The data you  state do not overwhelmingly  link the use of PPI to lower bone density and therefore increased risk of fracture. There may be a link to poor bone health and PPI use . Or if there is an increased risk of bone fracture with PPI use ,it could be  that use of PPIs is a marker for overall  frailty and  otherwise poor medical health and thus susceptibility to fracture.

 Specific line items:

Line 30:  if PPI drugs are over the counter, then they are not necessarily "prescribed"; people have access to them without prior medical evaluation. May want to change that term.

Line 32: you should provide your references #79-81 for this statement.

Line 81: ( and multiple other places) Does Vitamin B really only refer specifically to Vitamin B12,and  not vitamin  B1,B2,B3 or B6? Although these other B vitamins may be involved, you have only discussed data about  vitamin B12.

Line 299: You described data showing  increased osteoblastic viability which would lead to more bone, not less. This is an example of conflicting data, that you have  not mentioned in table 1.

Line 310:  You mention suppression of osteoclast formation which would lead to more bone not less bone , again conflicting with your diagram in table 1.

Line 318: where is the data about inhibition of osteoblast differentiation to support this statement as the prevalent mechanism and not just one of the conflicting studies (reference 101)?

Author Response

(The authors gave the same response as above.)

Reviewer 5 Report

The review article of “proton pump inhibitors and fracture risk: A review of current evidence and mechanisms involved” had performed a thorough review of the literature. Generally speaking, the anterior half of the article is written well and the posterior half of the article is a little chaos. I had some recommendation for this article as:

1.      Since different proton pump inhibitors had been mentioned in this article and may present different effects in their review, I suggest the author should introduce these proton pump inhibitors in the article for a better illustration to the readers.

2.      Throughout the article, most of the fracture risk indicated that occurred in the hip but some indicated in the other body regions. I think the author should indicate that in their description more clearly throughout the article to avoid confusion to the readers.

3.      In line 165-168, the lines 165-167 were redundant and could be deleted, the lines 167-168 said “Two main mechanisms centred on PPI-induced hypergastrinaemia and hypochlorhydria will be discussed” could be moved forwardly under the title of 3. Mechanism of bone fractures induced by PPI, because subtitle 3.1. was Hormones and PPIs, but the discussion about hypochlorhydria was put under the subtitle 3.2. Micronutrients and PPIs.

4.      In the figure 1, the author construct a link of “Serum Ca and Mg” to “hyperparathyroidism”, however, the discussion or referencing of this link could not be found in their article.

5.      Although the vitamin B (B12?) absorption is possible to be impaired with PPIs use and then cause muscle weakness and subsequent increased risk of falling, I do not appreciate the construction of such linkage as the complication of PPIs use and fracture risk. The reason is because that 1. the evidence only from very few articles that may have confounding effects in the assessment, 2. Some studies, as authors mentioned, did not support that PPI use would decrease Vitamin B level, and 3. the trauma mechanism is too complicated and the linkage of vitamin B absorption/demyelinating neurologic disease/muscle weakness/risk of falling to fracture sites was even not so evidenced and raised more question, such as what is the Vitamin B means here (B1, B2, B6, B12…)? Is the fracture associated with fall the hip fracture? Is the demyelination limited in the CNS or PNS systems? Is the patient with neurological disease that resulted from vitamin B deficiency loss their control and balance (but not only muscle weakness) and cause an increased fall risk? What is the increased odds of risk of fracture in different sites of the body in a ground level fall that caused by muscle weakness?

6.      The description under the title “future directions and conclusion” could be shorter and more concise.

Author Response

(The authors gave the same response as above.)

Round 2

Reviewer 3 Report

Dear authors,

Thank you for your response.

Reviewer 4 Report

corrections have improved paper